# RNAinsecta: A tool for prediction of precursor microRNA in insects and search for their target in the model organism *Drosophila melanogaster*

**Adhiraj Nath, Utpal Bora**[ID]*

Department of BSBE, IIT Guwahati, North Guwahati, Assam, India

* ubora@iitg.ac.in

**Data Availability Statement:** All files are available at: https://github.com/adhiraj141092/RNAinsecta

**Funding:** The authors received no specific funding for this work.

## Abstract

### Introduction and background

Pre-MicroRNAs are the hairpin loops from which microRNAs are produced that have been found to negatively regulate gene expression in several organisms. In insects, microRNAs participate in several biological processes including metamorphosis, reproduction, immune response, etc. Numerous tools have been designed in recent years to predict novel pre-microRNA using binary machine learning classifiers where prediction models are trained with true and pseudo pre-microRNA hairpin loops. Currently, there are no existing tool that is exclusively designed for insect pre-microRNA detection.

### Aim

Application of machine learning algorithms to develop an open source tool for prediction of novel precursor microRNA in insects and search for their miRNA targets in the model insect organism, *Drosophila melanogaster*.

### Methods

Machine learning algorithms such as Random Forest, Support Vector Machine, Logistic Regression and K-Nearest Neighbours were used to train insect true and false pre-micro-RNA features with 10-fold Cross Validation on SMOTE and Near-Miss datasets. miRNA targets IDs were collected from miRTarbase and their corresponding transcripts were collected from FlyBase. We used miRanda algorithm for the target searching.

### Results

In our experiment, SMOTE performed significantly better than Near-Miss for which it was used for modelling. We kept the best performing parameters after obtaining initial mean accuracy scores >90% of Cross Validation. The trained models on Support Vector Machine achieved accuracy of 92.19% while the Random Forest attained an accuracy of 80.28% on our validation dataset. These models are hosted online as web application called

**Competing interests:** The authors have declared that no competing interests exist.

RNAinsecta. Further, searching target for the predicted pre-microRNA in *Drosophila melanogaster* has been provided in RNAinsecta.

## Introduction

Pre-microRNA (pre-miRNA) are the precursor of microRNA (miRNA) from which one or more miRNAs are produced. microRNAs (miRNA) are a class of non-coding RNA which regulate gene expression. They are typically $\sim$22 bp long and bind to the 3′ untranslated region (3′ UTR) of target mRNAs to induce mRNA degradation and translational repression [1], however, recent studies have suggested that they also bind to the 5' UTR, coding region and gene promoters [2]. It was first discovered in *Caenorhabditis elegans* by Ambros and Ruvkun groups in 1993 [3, 4]. Since then it has been discovered in a large number of species across different kingdoms.

The role of miRNA is crucial in insects as it is reported to participate in a wide range of biological activities [5]. Changes in the miRNA profiles have been observed during metamorphosis where miR-100/let-7/miR-125 cluster has been found to participate in wing morphogenesis in both hemimetabolan and holometabolan species [6–8]. In reproduction, during ovarian development miR-309 plays a critical role in female *A. aegypti* mosquitoes and during spermatogenesis miR-7911c-5p is upregulated in *B. dorsalis* [9, 10]. Several miRNAs has been found to play important role in the regulation of immune related genes [11, 12] and also during insecticide resistance where the genes responsible are downregulated with the help of miRNAs, miR-2b-3p is found to be involved in regulation of metabolic resistance[13, 14].

In insects, the pre-miRNA hairpin is exported from the nucleus to the cytoplasm by Exportin-5, where it is further processed by the RNase III enzyme Dicer-1. In mammals, Dicer-1 processes pre-miRNAs into miRNA duplexes that are then loaded into the RNA-induced silencing complex (RISC).However, in insects, Dicer-1 processes pre-miRNAs into mature miRNAs directly, which are then loaded onto the Argonaute protein to form the functional RISC complex. The mature miRNA then guides the RISC complex to target mRNAs for translational repression or degradation [15–17].

Numerous tools have been designed to predict novel pre-miRNA using machine learning approaches by training data to classify pre-miRNA hairpin from pseudo pre-miRNA. Tools for species specific novel pre-miRNA detection like TamiRPred [18] and phylum specific such as ViralMir [19] have also been developed. Most of the tools use the characteristics of the hairpin loop as features for the classification [20–26]. Most tools consider 8,494 non-redundant human pseudo hairpins as the negative dataset [20, 21, 27–30], however selection of negative dataset still remains a challenge and careful consideration is required to make efficient binary supervised classification models [31, 32].

Genomic hairpin sequences which are not pre-miRNA viz., mRNA, tRNA and rRNA are also used as negative set [33]. However, inclusion of such collection of pseudo-hairpins give rise to the class-imbalance problem. This issue is addressed in tools like HuntMi, where thresholding classifier score function is combined with receiver operating characteristics (ROC) [34], microPred where the concept of undersampling majority class and oversampling minority class was used [35] and DeepSOM addresses this issue by creating self-organizing maps [36].

Tools have also been developed to search for potential miRNA target sites in a genomic sequence such as miRanda, Pictar, mirmap [37–39] etc. These tools search for potential target sites for a given sequence in a gene by calculating likelihood, allowing wobble basepairing and

reward and complementarity at 5' end. Recently tool for genome wide pre-miRNA detection, MiRe2e was also developed using deep learning model [40].

The pre-miRNA sequences of insects differ from human, plants and mouse in length, MFE, GC%, etc. upon which most of the available tools are trained on. As miRNA plays a major role in insects and yet a tool which is exclusively dedicated for its detection is not available, we have designed an ML based pre-miRNA prediction tool while handling the class imbalance problem using SMOTE. The pre-miRNA predicted as positive can also be used to search for probable targets in genes of *Drosophila melanogaster* chromosomes that have been reported to be regulated by miRNAs.

## Methods

### Data preprocessing for binary ml classification

**Data collection.** In order to make a binary classification, we prepared two datasets (categories/groups), positive and negative pre-miRNA. For preparing the positive pre-miRNA (true pre-miRNA) dataset, we downloaded all the available insect pre-miRNA sequences from miRBase [41]. A total of 3391 sequences were collected and labelled as positive set for the ML classification (available at: https://github.com/adhiraj141092/RNAinsecta/blob/master/dataset/insect_miRNA.fasta).

For the negative dataset (pseudo pre-miRNA), we initially used the 8494 pseudo-hairpin sequences implemented in developing previous tools in humans and other organisms [19, 27, 35]. We found this negative dataset to overfit in training insect ml binary classifiers. Hence, we enriched our negative set with genomic sequences of different insects Protein Coding Genes (PCGs). We downloaded more than 1,00,000 PCGs from GenBank using e-search and e-fetch APIs of e-utilities'[42]. We retained the sequences with length below 250bp since the longest pre-miRNA reported for insects is pxy-mir-8515-1, 222 bp long produced by *Plutella xylostella* (miRBase Accession: MI0027331). We then calculated the secondary structure and minimum free energy (MFE) using RNAfold of ViennaRNA package [43]. These datasets were further processed to extract the sequences and their corresponding secondary structure notation (dots and brackets for basepairing) and MFE value to tabular format using regex in in-house python script. We then filtered the sequences based on MFE from -5 to -180, since dvi-mir-315b in *Drosophila virilis* is found to have the highest MFE of -5.4 (miRBase Accession: MI0009499) and the same pre-miRNA from *Plutella xylostella* (miRBase Accession: MI0027331) was found to have lowest MFE of -174.9. GC content (%G+C) was calculated by in-house python script and we chose the sequences with GC between 10–85%, since the lowest GC content was found to be 12.28% in pxy-mir-8547 (miRBase Accession: MI0027419) and the highest in pxy-mir-8517a (miRBase Accession: MI0027332). After filtering, a total of 23,252 negative dataset sequences from insect PCGs and previous pseudo pre-miRNA was prepared. This dataset can be found in RNAinsecta GitHub repository (https://github.com/adhiraj141092/RNAinsecta/blob/master/dataset/pseudo_insect_pre-mIR.csv).

### Features for the binary ml classification

We calculated different measurable properties of both the classes (groups) and labelled them 0 and 1 based on which the ML models were trained. A total of 93 features were calculated as described below:

**Triplet element scores.** We used TripletSVM's method for calculating the triplet element scores where, given any three adjacent nucleotides, there are eight ($2^3$) possible structure compositions: '(((', '((.', '(..', '...', '.((', '..(', '.(.' and '(.(', taking '(' for both instances of paired nucleotide. Considering the middle nucleotide, there are 32 ($4 \times 8$) possible structure-sequence

combinations, which are denoted as 'C(((', 'G((.', etc. We used a perl script for the triplet calculation [20].

**Base composition.** The nucleotide and its percentage:

$\%X = \frac{|X|}{(L)} * 100$, where $X \in \{A, C, G, U\}$ and L = Length

dinucleotide counts and their percentage:

$\%XY = \frac{|XY|}{(L-1)} * 100$, where $X, Y \in \{A, C, G, U\}$ and L = Length

base pair composition:

$$\%(X + Y) = \frac{|X| + |Y|}{L} * 100, \text{ where L = Length and } X, Y = \begin{cases} X = C \text{ and } Y = G \\ \qquad or \\ X = A \text{ and } Y = U \end{cases}$$

**Structural and thermodynamic features.** Number of stems, loops, loop length and number of basepairs were calculated from the secondary structure using regular expression and were used as features. A motif containing more than three contiguous base pairs in the secondary structure is termed as stem. The features dG, dP, dD, dQ, normalized Shannon entropy, MFE1 and MFE2 were adapted from miPred perl script [27].

dG is calculated by taking the ratio of MFE to the Length i.e. *dG = MFE / L*. Normalized base-pairing propensity, $dP = \frac{tot_{bp}}{L}$, where $tot_{bp}$ is the total basepairs and $L$ is the Length. MFE1 is the ratio between dG and GC content, i.e. *MFE1 = dG / (%G + C)* and MFE2 is the ratio between dG and number of stems, i.e. *MFE2 = dG / n_stems*, where *n_stems* is a structural motif containing more than three contiguous base pairs. All these features were calculated using in-house python script.

MFE3 and MFE4 features were implemented from microPred [35]. MFE3 is the ratio between dG and number of loops, i.e. *MFE3 = dG / n_loops*, where *n _loops* is the number of loops in the secondary structure. MFE4 is the ratio between dG and the total number of bases i.e. *MFE3 = dG / tot_bases* where *tot_bases* is the total number of base pairs in the secondary structure. dD is the adjusted basepair distance and zD is normalized dD. Normalized Shannon entropy is given by $dQ = -\sum_{i=1}^{j} \frac{(p_{ij}) \cdot \log_2 p_{ij}}{L}$, where the probability that base *i* pair with base *j* is then given by $p_{ij}$ and L is the Length [44]. Average basepair was calculated by taking the ratio of total bases and *n_stems*, i.e. *avg_bp = tot_bases / n_stems*.

**Handling class imbalance.** As there is a huge difference in ratio of positive to negative classes, state of the art techniques were implemented to address class imbalance. Two strategies namely Synthetic Minority Over-sampling Technique (SMOTE) [45] and Near-Miss (NM) [46, 47] were used to balance the dataset. Packages in python are available for implementation of both techniques [48].

## Classification and performance evaluation

**Classification algorithms.** The training was performed on different ML algorithms viz., Support Vector Machine (SVM), Random Forest (RF), Logistic Regression (LR) and k-Nearest Neighbours (KNN) to classify the positive and negative labelled miRNA from the calculated features. The dataset was divided into a training set (X_train) which consisted of 75% of the data and a testing set which consisted of 25% (X_test) of the data [49].

**Hyperparameter tuning.** Different parameters of the ML algorithms were applied to the SMOTE, NM and unbalanced datasets to classify the positive and negative miRNA. For SVM, we used linear kernel: $K\left(x_i, x_j\right) = x_i^T x_j$, polynomial kernel: $K\left(x_i, x_j\right) = \left(\gamma x_i^T x_j + r\right)^d$ and

radial basis function (RBF) kernel: $K\left(x_i, x_j\right) = \exp\left\{-\gamma\lVert x_i - x_j\rVert^2\right\}\}$, where $x \in R$, (i = 1,2,3..,
N) are inputs and $\gamma, r, d > 0$ are kernel parameters. Different values of the Cost function
($C_{SVM}$ value) and gamma were adjusted in the SVM algorithm optimization.

In the case of RF the model works on: $RFfi_i = \dfrac{\sum_{j \in N} norm\, fi_{ij}}{T}$, where $RFfi_i$ is the importance
of feature $i$ ($fi_i$) calculated from N which denotes all trees in the Random Forest model, $normfi_{ij}$
is the normalized feature importance for i in tree j, i.e. $norm\, fi_i = \dfrac{fi_i}{\sum_{j \in P} fi_j}$ and T is total num-
ber of trees and F denotes all features. We henceforth, chose different values for the number of
trees, learning rate, maximum depth, minimum number of sample split and sample leaf were
used.

For kNN, number of neighbours and different distance matrices were used such as
$euclidean = \left\{\sum_{i=1}^{k} (x_{1i} - x_{2i})^2\right\}^{\frac{1}{2}}$, $manhattan = \sum_{i=1}^{k} |x_{1i} - x_{2i}|$, $minkowski$
$= \left\{\sum_{i=1}^{k} |x_{1i} - x_{2i}|^p\right\}^{\frac{1}{p}}$, where k is the number of neighbours to be considered for calculating
the distance and $x \in R$, (i = 1,2,3..,N) are inputs.

The logistic regression algorithm works on: $LR = \dfrac{e^{(\beta_0 + \beta_1 x)}}{1 + e^{(\beta_0 + \beta_1 x)}}$ where $\beta_0 + \beta_1 x$ is the equation of
straight line with $\beta_1 x$ as slope and $\beta_0$ as y-intercept which is converted to natural log. Regulari-
zation strength ($C_{LR}$ value) similar to cost function of SVM provides a penalty score and differ-
ent solvers were used to optimize the LR algorithm.

We used python's scikit-learn package to choose the hyperparameters for training each
algorithm [50]. Initially, we chose a wide range of hyperparameters for each of the above-men-
tioned parameters and classified using a model selection package called RandomizedsearchCV
which randomly chooses different parameters to train the ML algorithm with 10-fold cross val-
idation. We then fine-tuned the parameters using GridsearchCV, where the training was per-
formed using each of the possible combinations of the provided parameters along with 10-fold
Cross Validation (CV).

10-Fold CV essentially splits the dataset into 10 parts and trains 9 parts with a given param-
eter and use 1 to test the data. This repeats for all 10 parts and the mean accuracy score is pro-
vided as CV score. For example, in case of RF, we used No. estimators: 10 to 5000, Max depth:
5 to 100, Bootstrap: True and False, Min sample leaf: 1 to 10, Min sample split: 1 to 10. For
SVM we used, Cost Function ($C_{svm}$): 0.25 to 100, Kernel: Linear, Polynomial and RBF,
Gamma: 0 to 10. For KNN, No. of neighbours between 1 and 50, distance metrics: Euclidean,
manhattan and minkowski were used. In case of LR, we used cost function ($C_{LR}$): 10 to 100.
RandomSearchCV initially selects random parameter values from the range and performs a
10-fold training resulting in mean accuracy score CV for a given algorithm. Then using Grid-
SearchCV, exact parameter value was provided from the range of obtained values from Ran-
domSearchCV. In GridSearchCV, one optimum parameter value once reached is fixed and the
remaining parameter is optimized one at a time, resulting in the fine-tuning of the parameters
for the ml classification model.

**Test set.** Initially X_test was used to evaluate performance for all the classifiers. The
X_test consisted of 853 positive and 5818 negative entries. Further, to generalize the model, an
independent test dataset was created from the pre-miRNA sequences of the insects *Spodoptera
frugiperda* [51] and *Tribolium castaneum* [52, 53] which were not used in the initial data col-
lection and hence remained entirely unseen to the project, were considered as positive dataset.
Insect CDS of 250bp length fetched from GenBank using the same steps as mentioned above
in "Data Collection" were considered as negative dataset. A total of 999 sequences were

considered as the validation dataset (V_test) of which 464 were positive and 535 were negative as given in the GitHub link (https://github.com/adhiraj141092/RNAinsecta/tree/master/dataset/pos.fold and https://github.com/adhiraj141092/RNAinsecta/tree/master/dataset/neg.fold respectively). We further evaluated the model with imbalance data test (M_test) using the same 464 positive and enriching the negative set to 116,230 entries which were created using combination of naturally occurring and artificially generated dataset to mimic ncRNA that closely resemble with pre-miRNA instead of long mRNA that are bound to be TN. By doing this, we aimed to minimize the risk of overfitting.

**Performance evaluation.** The best performing models for each classifier were selected for SMOTE, NM and imbalance dataset after adjusting the hyperparameters. The Cross-Validation score (CV Score), which is the mean accuracy of the 10 folds, was considered for performance and the parameters yielding the highest mean accuracy were selected for each classifier. The best parameters for each classification algorithm were chosen and the models were evaluated on X_test dataset to check for overfitting during the training. Their performance was calculated based on the following classical classification measures: sensitivity (SN): $SN = \frac{TP}{TP+FN}$, specificity (SP): $SP = \frac{TN}{TN+FP}$, Accuracy (Acc): $Acc = \frac{TN+TP}{TN+FP+TP+FN}$, precision (p): $p = \frac{TP}{TP+FP}$, harmonic mean of sensitivity and precision ($F_1$): $F_1 = 2\frac{SN \cdot p}{SN+p}$ and Matthew's correlation coefficient (MCC): $MCC = \frac{(TP \cdot TN)+(FP \cdot FN)}{\sqrt{(TP+FP)(TP+FN)(TN+FP)(TN+FN)}}$, where TP, TN, FP and FN are the number of true-positive, true-negative, false-positive and false-negative classifications, respectively. For given false positive rate (α) and true positive rate (1 − β) at different threshold values, the AUC-ROC was computed as: $AUC = \sum_{n=1}^{i}\left\{(1 - \beta_i\Delta\alpha) + \frac{1}{2}[\Delta(1 - \beta)\Delta\alpha]\right\}$, where Δ(1−β) = (1 −β$_i$)−(1−β$_{i−1}$) and Δα = α$_i$−α$_{i−1}$ and i = 1, 2, . . ., m (number of test data points) [18]. In imbalance class testing data, accuracy, sensitivity, specificity, precision and $F_1$ are not the best measures to analyse performance of models as they are not based on the entire confusion matrix. MCC is a better estimator of performance in such cases as it produces a high score only if good results are obtained in all of the four confusion matrix categories [54].

test.py (https://github.com/adhiraj141092/RNAinsecta/blob/master/test.py) can be used to replicate the results.

## Web development for target searching

**Data collection and annotation.** We initially downloaded the genome coordinates of *Drosophila melanogaster* pre-miRNA from miRBase (https://www.mirbase.org/ftp/CURRENT/genomes/dme.gff3). Experimentally verified miRNA and their target gene IDs of *Drosophila melanogaster* were fetched from MirTarBase [55] which was used to extract the relevant IDs from the genome coordinates obtained from miRBase. Parent IDs were annotated from the target gene ID list using e-utilities [42], with which the CDS of the genes were downloaded from Flybase [56]. We used regular expression in e-utilities to match the patterns and retrieve the IDs since single miRNA regulates multiple transcripts. A total of 174 target transcripts were collected and stored.

**Web server implementation.** The selected trained models were implemented in a backend server using python's Flask API on a cloud platform along with NGINX [57] as reverse proxy as given in Fig 1. Input from the user is received by NGINX as an HTTP request which it sends to the backend Flask server using reverse proxy. The request from NGINX is interpreted by the Flask API using Gunicorn which is a python WSGI (Web Server Gateway Interface) HTTP server [58]. A port was assigned to the Flask process by Gunicorn with which NGINX communicates. Gunicorn is run in background using Supervisor which also monitors

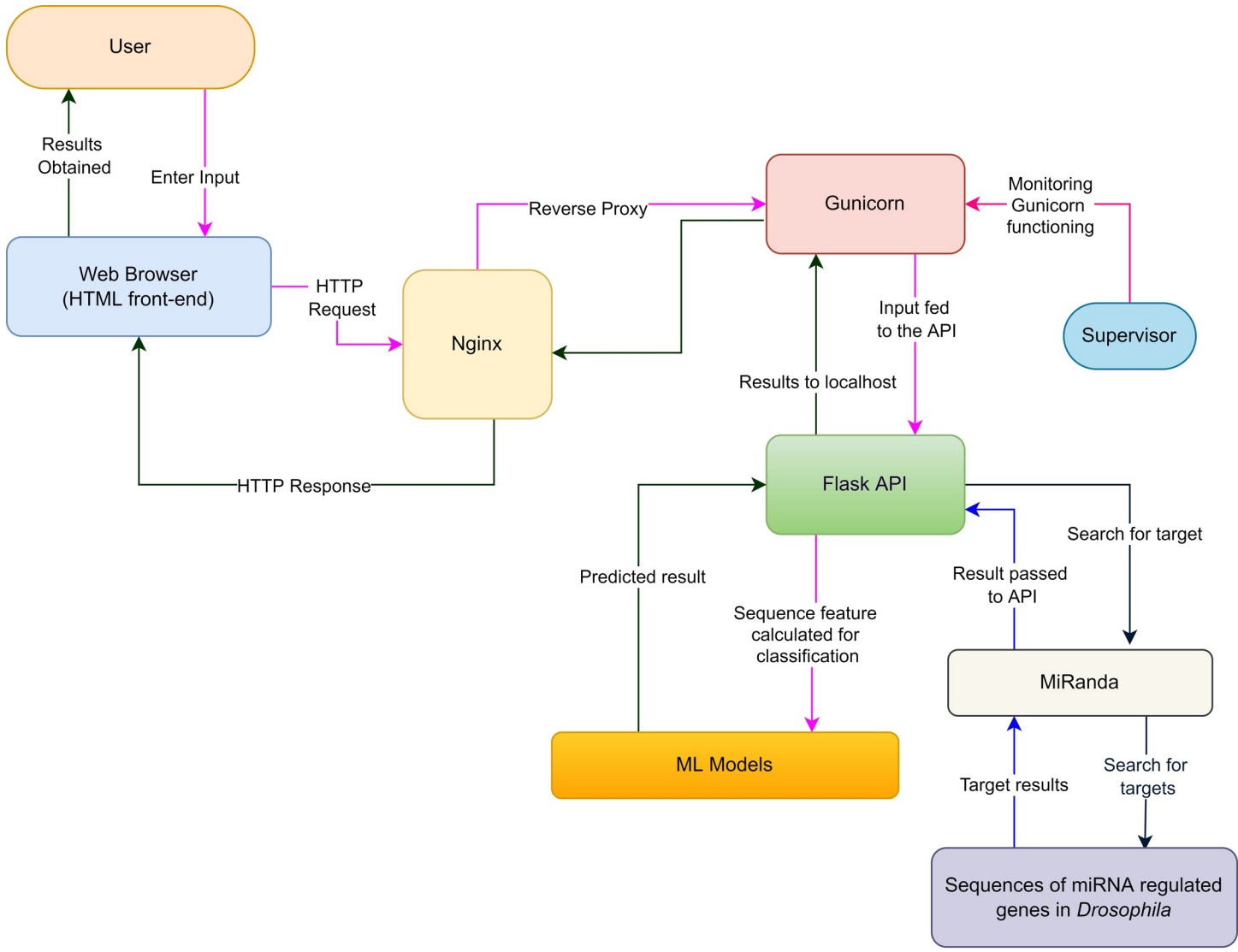

**Fig 1. Web-server implementation of RNAinsecta.** The figure shows the full-stack implementation of the website. NGINX takes nucleotide sequence information as HTTP request from User through the Homepage. Flask runs the API for pre-miRNA and target prediction as a localhost. Gunicorn works as mediator between NGINX and Flask which allows public IP to interact with the APIs. The monitoring of Gunicorn is done by supervisor, which prepares error reports and restarts the server if it stops unexpectedly.

the process, keeps track of errors and restarts the app in case it stops. In the Flask app the sequence features are calculated and the selected ML model predicts if the given sequences are pre-microRNAs or not. The results are sent to the browser from the Flask app as Jinja2 templates [59]. HTML, CSS, Javascript and JQuery were used along with Jinja2 template for the frontend design. The website loads securely with SSL certificate generated by certbot which ensures no malicious activity being done [60].

We have further implemented miRanda [37], miRmap [39] and RNAhybrid [61] to enable users to search for the potential miRNA targets for their pre-miRNA in the transcript of reported genes regulated by miRNA.

## Results

### Data preprocessing

**Datasets.** The number of true pre-miRNA sequences obtained for each species from miR-Base is given in Table 1. Insect pre-miRNA differs from other organisms in various aspects as given in S1 File, for which the previous tools do not perform well classifying insect pre-miRNA sequences.

**Handling class imbalance.** The SMOTE training dataset consisted of 17,475 instances for both the classes making it a total of 34,950 instances. In case of NM training set, both the classes had 2920 instances making it a total of 5,840 instances.

### Feature selection

The f score and $p$-value of all 93 features are given in S2 File. The features MFE3 and MFE4 had the highest f score value which suggests that the data points for these two features differ the most between positive and negative datasets. We also used standard scaler for normalizing from preprocessing package of sklearn [50].

**Table 1. The number of microRNA sequences collected from each species for building the machine learning model.**

| Organism | No. of precursors |
|---|---|
| *Bombyx mori* | 487 |
| *Drosophila melanogaster* | 258 |
| *Apis mellifera* | 254 |
| *Drosophila pseudoobscura* | 210 |
| *Drosophila virilis* | 180 |
| *Aedes aegypti* | 155 |
| *Drosophila simulans* | 148 |
| *Plutella xylostella* | 133 |
| *Anopheles gambiae* | 130 |
| *Acyrthosiphon pisum* | 123 |
| *Drosophila sechellia* | 103 |
| *Dinoponera quadriceps* | 102 |
| *Drosophila erecta* | 101 |
| *Manduca sexta* | 98 |
| *Heliconius Melpomene* | 92 |
| *Drosophila yakuba* | 89 |
| *Drosophila grimshawi* | 82 |
| *Bactrocera dorsalis* | 80 |
| *Drosophila willistoni* | 77 |
| *Drosophila ananassae* | 76 |
| *Drosophila persimilis* | 75 |
| *Culex quinquefasciatus* | 74 |
| *Polistes canadensis* | 73 |
| *Drosophila mojavensis* | 71 |
| *Nasonia vitripennis* | 53 |
| *Nasonia giraulti* | 32 |
| *Nasonia longicornis* | 28 |
| *Locusta migratoria* | 7 |

## Classification and performance evaluation

**Selection of parameters.** The initial parameters were selected based on the best performing models for SMOTE, NM and imbalance dataset. The overall CV score of NM was found to be lower than the SMOTE dataset. The best parameters for all the classifiers are given in Table 2 which was used in final model preparation and evaluation.

In case of SVM, we found the cost function $C_{SVM}$ to range from 1 to 10 for optimum classification. SMOTE used $C_{SVM}$ 10 while NM and Imbalance set used 1 and 6 respectively. All datasets performed best with RBF kernel with Gamma 0.01 for SMOTE while 0.001 for NM and Imbalance set. The 10-Fold CV score was found to be 0.9946 in SMOTE, 0.9528 in NM and 0.9687 in case of Imbalanced set.

RF classifiers gave best performance with the value of minimum sample leaf and sample split fixed at 1 and 2 respectively while the remaining parameters were further tuned. Maximum depth of 30 was found to be optimal for modelling and hence was assigned in both SMOTE and NM datasets whereas in case of Imbalance dataset, maximum depth of 10 was found to be optimum. The number for estimators yielding best performance for SMOTE was 1600 which was slightly more than NM which had 1400 estimators, whereas 30 estimators were found to be optimum in case of Imbalance dataset. The 10-Fold CV score was found to be 0.9801 in SMOTE, 0.9446 in NM and 0.9628 in case of Imbalanced set.

Logistic Regression classifiers performed poorly on SAG and SAGA solvers. The performance was slightly better with Liblinear, however the best performance was obtained with Newton-CG solver. $C_{LR}$ value was tuned after fixing the solver and a value of 70, 100, 30 were found to be optimum for SMOTE, NM and Imbalance dataset respectively. The 10-Fold CV score was found to be 0.9627 in SMOTE, 0.9470 in NM and 0.9671 in case of Imbalanced set.

KNN classifiers used either Manhattan or Euclidean as distance metric. The performance was greatly influenced by the number of neighbours as expected. 2 neighbours were found to be optimum in case of SMOTE while 7 neighbours were used for NM and Imbalance dataset. The 10-Fold CV score was found to be 0.9820 in SMOTE, 0.9215 in NM and 0.9603 in case of Imbalanced set. The details of the CV scores for each fold during the training process can be found in S3 File.

**Performance evaluation.** Each trained model obtained from SMOTE, NM and imbalance dataset was tested on the same X_test dataset with no artificial balancing to overcome the

**Table 2. Sets of best performing parameters obtained after gridsearching through different values for each classification algorithm.** Selection was based on the highest 10-fold cross-validation score. Parameters tuned for SVM were Cost function $C_{SVM}$, Kernel and Gamma value. Parameters considered for RF were No. estimators, Max depth, Min sample leaf and Min sample split. For LogR the parameters considered were Cost Function $C_{LR}$ and solver. For KNN, No. of neighbours and Metric Distance was considered for tuning. The CV score shows the mean accuracy score of the best classifier obtained with the corresponding parameters.

| Classifier | SMOTE Best CV Score | SMOTE Best Parameters | NM Best CV Score | NM Best Parameters | Imbalance Best CV Score | Imbalance Best Parameters |
|---|---|---|---|---|---|---|
| SVM | 0.99459 | $C_{SVM}$ = 10, Kernel = : RBF, Gamma = 0.01 | 0.952789 | $C_{SVM}$ = 1, Kernel = : RBF, Gamma = 0.001 | 0.96871 | $C_{SVM}$ = 6, Kernel = : RBF, Gamma = 0.001 |
| RF | 0.98609 | No. estimators = 1600 Max depth = 30, Bootstrap = 'False', Min sample leaf = 1, Min sample split = 2, | 0.944617 | No. estimators = 1400 Max depth = 30, Bootstrap = 'False', Min sample leaf = 1, Min sample split = 2, | 0.96281 | No. estimators = 30 Max depth = 10, Bootstrap = 'False', Min sample leaf = 2, Min sample split = 2, |
| LogR | 0.9627 | $C_{LR}$ = 70, Solver = Newton-cg | 0.946952 | $C_{LR}$ = 100, Solver = Newton-cg | 0.96703 | $C_{LR}$ = 30, Solver = Newton-cg |
| KNN | 0.98199 | Metric Distance: Manhattan, No. of neighbours = 2 | 0.92166 | Metric Distance = Euclidean, No. of neighbours = 7 | 0.96032 | Metric Distance = Euclidean, No. of neighbours = 7 |

sampling bias in their comparison. The selection of such large datasets for testing gives a better understanding of their performance. Table 3 contains the performance measures for all the selected classifiers. Sl. No. 1–4 contains the performance for SMOTE dataset classification in SVM, RF, LogR and KNN respectively. SVM and RF had the highest accuracy of 0.9745 and 0.9835 respectively in SMOTE dataset followed by KNN and LogR with 0.9707 and 9695 respectively. The performance for NM classifiers is given in Sl. No. 5–8. The accuracy score for SVM, RF, LogR and KNN for NM classifier are 0.4218, 0.4368, 0.4465 and 0.4312 respectively. Sl. No. 9–12 contains the performance for Imbalance dataset. The accuracy score for SVM, RF, LogR and KNN for NM classifier are 0.9834, 0.9783, 0.9754 and 0.9767 respectively. We also provide the performance for the 8494 human_CDS negative dataset that is used by many previous tools as negative class for our insect pre-miRNA classification which is given in Sl. No. 5–8. The accuracy score for SVM, RF, LogR and KNN for NM classifier are 0.4383, 0.1377, 0.4336 and 0.2643 respectively.

The MCC scores for SMOTE: SVM, RF, LogR and KNN was found to be 0.9053, 0.9340, 0.8882 and 0.8854 respectively. NM dataset's MCC for SVM, RF, LogR and KNN was found to be 0.271, 0.2811, 0.2848 and 0.2858 respectively. The MCC scores for Imbalance dataset: SVM, RF, LogR and KNN was found to be 0.9289, 0.9066, 0.8941 and 0.9001 respectively. The MCC score for human_CDS negative dataset performance was found to be 0.2109, 0.00, 0.1991 and 0.1522 respectively.

**Comparision with previous tools using V_test.** The SMOTE and imbalance models were further analysed for their performance in comparison to the already developed tools, viz., miPred, microPred, Triplet-SVM, HuntMi and MiPred on the same validation dataset, V_test, comprising of 464 positive and 536 negative sequences. Table 4 consists of the performance measures for each of the tools along with the imbalance set. From Sl. Nos. 1 to 5, the performance for the previous tools is provided. Triplet-SVM had an accuracy of 0.7548, MiPred had an accuracy of 0.4805, microPred had an accuracy of 0.7788, HuntMI had an accuracy of 0.6186. Triplet-SVM's MCC was 0.6257, HuntMI's was 0.4141, MiPred's was 0.3256, miPred's was 0.6541, and microPred's was 0.5743. Triplet- SVM's specificity and sensitivity were 0.7981

**Table 3. Performance measure for each classifier of SMOTE from Sl. No. 1–4, NM from Sl. No. 5–8, Imbalance from Sl. No. 9–12 and 8494 human_CDS negative datasets from Sl. No. 13–16.** Accuracy, Specificity (SP), Sensitivity (SN), Matthew's correlation coefficient (MCC), Precision (p), harmonic mean of sensitivity and precision ($F_1$) are given corresponding to each ML classifier.

| Sl. No. | Classifier | Acc | SP | SN | MCC | P | $F_1$ |
|---|---|---|---|---|---|---|---|
| 1 | SVM_SMOTE | 0.97451 | 0.975685 | 0.967611 | 0.90525 | 0.871468 | 0.917026 |
| 2 | RF_SMOTE | 0.983498 | 0.992757 | 0.92915 | 0.934044 | 0.95625 | 0.942505 |
| 3 | LogR_SMOTE | 0.969501 | 0.971029 | 0.960526 | 0.8882 | 0.849597 | 0.901663 |
| 4 | KNN_SMOTE | 0.970679 | 0.983101 | 0.897773 | 0.885446 | 0.900508 | 0.899138 |
| 5 | SVM_NM | 0.421836 | 0.331954 | 0.949393 | 0.270949 | 0.194929 | 0.323448 |
| 6 | RF_NM | 0.436865 | 0.35075 | 0.942308 | 0.281079 | 0.198254 | 0.327586 |
| 7 | LogR_NM | 0.446589 | 0.361097 | 0.948381 | 0.284791 | 0.201853 | 0.33286 |
| 8 | KNN_NM | 0.431266 | 0.349026 | 0.913968 | 0.285773 | 0.193031 | 0.318743 |
| 9 | SVM_imbalanced | 0.983351 | 0.993848 | 0.917647 | 0.928914 | 0.959732 | 0.938218 |
| 10 | RF_imbalance | 0.978341 | 0.994874 | 0.874866 | 0.906552 | 0.964623 | 0.917555 |
| 11 | LogR _imbalance | 0.975394 | 0.991285 | 0.875936 | 0.894082 | 0.941379 | 0.907479 |
| 12 | KNN_imbalance | 0.97672 | 0.991114 | 0.886631 | 0.900102 | 0.940976 | 0.912996 |
| 13 | SVM_human_CDS | 0.438338 | 0.36056 | 0.925134 | 0.210874 | 0.187758 | 0.312162 |
| 14 | RF_human_CDS | 0.137763 | 0 | 1 | 0 | 0.137763 | 0.242165 |
| 15 | LogR_human_CDS | 0.433623 | 0.357143 | 0.912299 | 0.199071 | 0.184832 | 0.307387 |
| 16 | KNN _human_CDS | 0.264329 | 0.146787 | 1 | 0.152158 | 0.157726 | 0.272476 |

**Table 4. Comparative performance analysis between available tools and trained models tested upon independent insect pre-miRNA validation dataset.** 1–5 shows the performance of previous tools. 6–9 shows the performance on the SMOTE classifiers, 10–13 shows the performance on Imbalance set. The parameters for evaluation are Accuracy, Specificity (SP), Sensitivity (SN), Matthew's correlation coefficient (MCC Precision (p), harmonic mean of sensitivity and precision ($F_1$) are given for each corresponding ML classifier.

| Sl. No. | Tool | Acc | SP | SN | MCC | P | $F_1$ |
|---------|------|-----|----|----|-----|---|-------|
| 1 | Triplet-SVM | 0.754755 | 0.798131 | 0.704741 | 0.625745 | 0.751724 | 0.727475 |
| 2 | MiPred | 0.48048 | 0.128972 | 0.885776 | 0.325557 | 0.468643 | 0.612975 |
| 3 | miPred | 0.778779 | 0.785047 | 0.771552 | 0.654075 | 0.756871 | 0.764141 |
| 4 | microPred | 0.713714 | 0.570093 | 0.87931 | 0.574294 | 0.639498 | 0.740472 |
| 5 | HuntMI | 0.618619 | 0.306542 | 0.978448 | 0.414076 | 0.550303 | 0.704422 |
| 6 | SVM_SMOTE | 0.921922 | 0.945794 | 0.894397 | 0.854779 | 0.934685 | 0.914097 |
| 7 | RF_SMOTE | 0.802803 | 0.695327 | 0.926724 | 0.676982 | 0.725126 | 0.813623 |
| 8 | LogR_SMOTE | 0.677678 | 0.450467 | 0.939655 | 0.513201 | 0.59726 | 0.730318 |
| 9 | KNN_SMOTE | 0.761762 | 0.583178 | 0.967672 | 0.614122 | 0.668155 | 0.790493 |
| 10 | SVM_imbalance | 0.535536 | 1 | 0 | 0 | 0 | 0 |
| 11 | RF_imbalance | 0.72973 | 0.857944 | 0.581897 | 0.461038 | 0.780347 | 0.666667 |
| 12 | LogR_imbalance | 0.575576 | 0.91028 | 0.189655 | 0.145339 | 0.647059 | 0.293333 |
| 13 | KNN_imbalance | 0.526527 | 0.983178 | 0 | 0 | 0 | 0 |

and 0.7047, MiPred's were 0.8858 and 0.7851, MicroPred's were 0.5701 and 0.8793, and Hunt-MI's were 0.3065 and 0.9784 correspondingly.

In Sl. No. 6–9., the SMOTE classifiers are listed. SVM_SMOTE accuracy was 0.9219, RF_SMOTE accuracy was 0.8028, LogR_SMOTE accuracy was 0.6777, and KNN_SMOTE accuracy was 0.7618. SVM_SMOTE, RF_SMOTE, LogR_SMOTE, and KNN_SMOTE all had MCC scores of 0.8548, 0.6770, 0.5132 and 0.6141, respectively. The specificity and sensitivity of SVM_SMOTE, RF_SMOTE, LogR_SMOTE, and KNN_SMOTE were each 0.9458 and 0.8944, 0.4505 and 0.9397, and 0.5832 and 0.9677, respectively.

From Sl. Nos. 10 to 13, the performance for the Imbalance dataset is provided. SVM_imbalance, RF_imbalance, LogR_imbalance, and KNN_imbalance all have accuracy values of 0.5355, 0.7297, 0.5756 and 0.5265, respectively. For SVM_imbalance, RF_imbalance, LogR_imbalance, and KNN_imbalance, the Specificity and Sensitivity were 0 and 1, 0.8579 and 0.5819, 0.9103 and 0.1897, and 0.9832 and 0, respectively. MCC for SVM_imbalance was 0, RF_imbalance was 0.4611, LogR_imbalance was 0.1453, and KNN_imbalance was also 0.

**ROC.** The ROC curve (Receiver Operating Characteristic curve) is used to measure the performance of a model at different classification thresholds. It is a plot between True Positive Rate (Sensitivity) and False Positive Rate (1 - Specificity). Higher the AUC (Area Under the Curve) of ROC, the better is the model at classifying, i.e. higher degree of separability. The ROC-AUC of the models is given in Fig 2. We kept the classifiers RF and SVM for this analysis as they performed better than the others and named it "RNAinseceta_RF" and "RNAinsecta_SVM" respectively. The tools used in performance evaluation gave binary output without probability values and to maintain uniformity we used the same, due to which the plot appears to be straight line as used in other similar studies [18]. Triplet-SVM had an AUC of 0.751; miPred, 0.778; microPred, 0.655; Mipred, 0.507; and HuntMi, 0.642. Both RNAinsecta_RF and RNAinsecta_SVM had AUCs of 0.834 and 0.920.

**Imbalance set performance.** We further tested our RNAinsectaRF and RNAinsectaSVM with imbalance dataset (M_test) to measure their performance on large imbalanced data.The negative class of the dataset was constructed with ncRNA since mRNAs are significantly longer which would overfit the data with a large number of TN. The result is given in Table 5. Out of

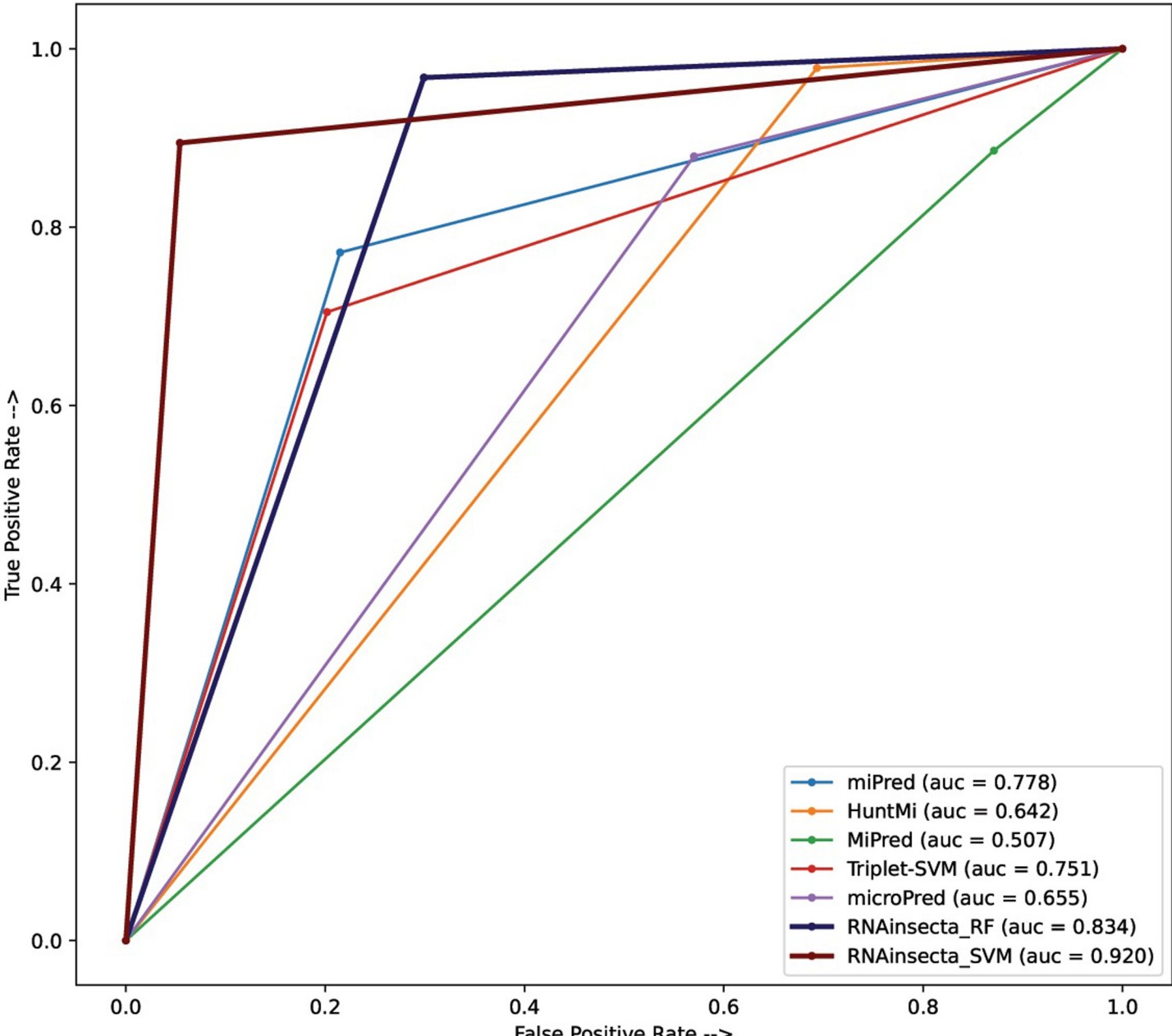

**Fig 2. ROC-AUC for comparative performance analysis of RNAinsecta with available tools for detection of insects' pre-miRNA.** The validation dataset used for this figure contains 464 positive and 536 negative sequences. Y axis contains the True Positive Rate (TPR) and X axis contains the False Positive Rate (FPR). More the AUC (Area Under the Curve) better is the performance.

**Table 5. Performance evaluation on imbalance class dataset (M_test) containing 116230 negative and 464 positive samples.**

| Classifier | TN | FP | FN | TP | acc | SP | SN | MCC | p | F1 |
|---|---|---|---|---|---|---|---|---|---|---|
| RNAinsecta_RF | 94136 | 22094 | 15 | 449 | 0.810539 | 0.809911 | 0.967672 | 0.12395 | 0.019917 | 0.039032 |
| RNAinsecta_SVM | 111147 | 5083 | 49 | 415 | 0.956022 | 0.956268 | 0.894397 | 0.252656 | 0.075482 | 0.139215 |

116,230 negative sequences 94,136 and 111,147 were correctly classified by RNAinsectaRF and RNAinsectaSVM respectively.

The accuracy, sensitivity, and specificity of RNAinsectaRF were 81%, 96.77%, and 81%, respectively. The accuracy, sensitivity, and specificity of RNAinsectaSVM were 95.6%, 89.43%, and 95.62%, respectively. However, the MCC, precision and recall for RNAinsectaSVM were 0.2526, 0.075482 and 0.139215 respectively. The MCC, precision and recall for RNAinsectaRF were 0.12395, 0.019917 and 0.039032 respectively. AUPRC (Area Under Precision Recall Curve) is considered optimal for evaluating binomial classification with imbalance class data-set [62]. The AUPRC which is bound to be less as there is a huge difference in the testing set for both the classes. The AUPRC plot is given in Fig 3. The AUPRC of SVM and RF model were found to be 0.62 and 0.08 respectively. This is primarily due to poor F1 and precision value as the negative sample is huge. This suggest it is not suitable for RNA-Seq pipeline yet.

**Performance on related phyla.** The performance of RNAinsecta was measured across species from other phyla and compared with miPred which has so far been better than other tools in our analysis. Their comparative performance is given in Table 6. pre-miRNA of various species from Nematoda, Platyhelminthes, Virus and Mollusca were taken.

In Platyhelminthes for instance, out of 148 pre-miRNAs from *Schmidtea mediterranea* RNAinsecta_RF correctly predicted 126 with a sensitivity 0.8514 while miPred correctly predicted 114 with a sensitivity of 0.7703. In *Gyrodactylus salaris*, out of 60 pre-miRNAs RNAinsecta_RF correctly predicted 52 with a sensitivity of 0.8667 while miPred identified 43 with a sensitivity of 0.7166.

In Nematoda, RNAinsecta_RF correctly predicted 208 of the 214 pre-miRNA from *Caenorhabditis brenneri* with a sensitivity of 0.9719 while miPred correctly predicted 194 with sensitivity of 0.9065. In *Brugia malayi*, out of 157, RNAinsecta_RF correctly predicted 122 with sensitivity of 0.7770 while miPred correctly predicted 119 with sensitivity of 0.7579.

In Virus such as *Duck enteritis*, out of 24 sequences RNAinsecta_RF correctly predicted 19 while miPred predicted 13. In Mollusca such as *Melibe leonina* out of 90 RNAinsecta_RF correctly predicted 83 sequences while miPred identified 85 sequences correctly.

## Web application and miRNA targets

We have listed the chromosome-wise miRNA target transcript distribution of *Drosophila melanogaster* in Table 7. There are total 176 target transcripts of which chromosome 2 Left and Right (2L and 2R) has 34 and 7 respectively. There are 46 targets for left and 55 for right of Chromosome 3 (3L and 3R) respectively. Chromosome 4 has 9 targets whereas sex chromosome X has 23 targets.

The web interface of RNAinsecta is given in Fig 4. It contains both RNAinsecta_RF and RNAinsecta_SVM classifiers with batch and single sequence query (Fig 4A). The exceptions for other inputs and empty query were handled at both front and back end. The result displays the prediction result and probability score for both batch and single sequence query (Fig 4B). In case of batch query, the HTML elements are dynamically created depending on the number of submitted query which is achieved using JQuery (Fig 4B.i). In case of single sequence query, user can download the secondary structure predicted for their pre-miRNA by RNAfold. Also, users can visualize various parameters of their sequence such as nucleotide and dinucleotide counts, and Triplet folding pattern which is achieved using chart.js JavaScript library (Fig 4B.ii).

For a particular single sequence query predicted to be true pre-miRNA, users can search for the mature miRNA targets in each chromosome given in Table 7 by preprocessing the pre-miRNA to putative miRNA. miRanda, miRmap and RNAhybrid programs are available in the back-end which searches the targets and displays the top 100 results (Fig 4C). The selection of

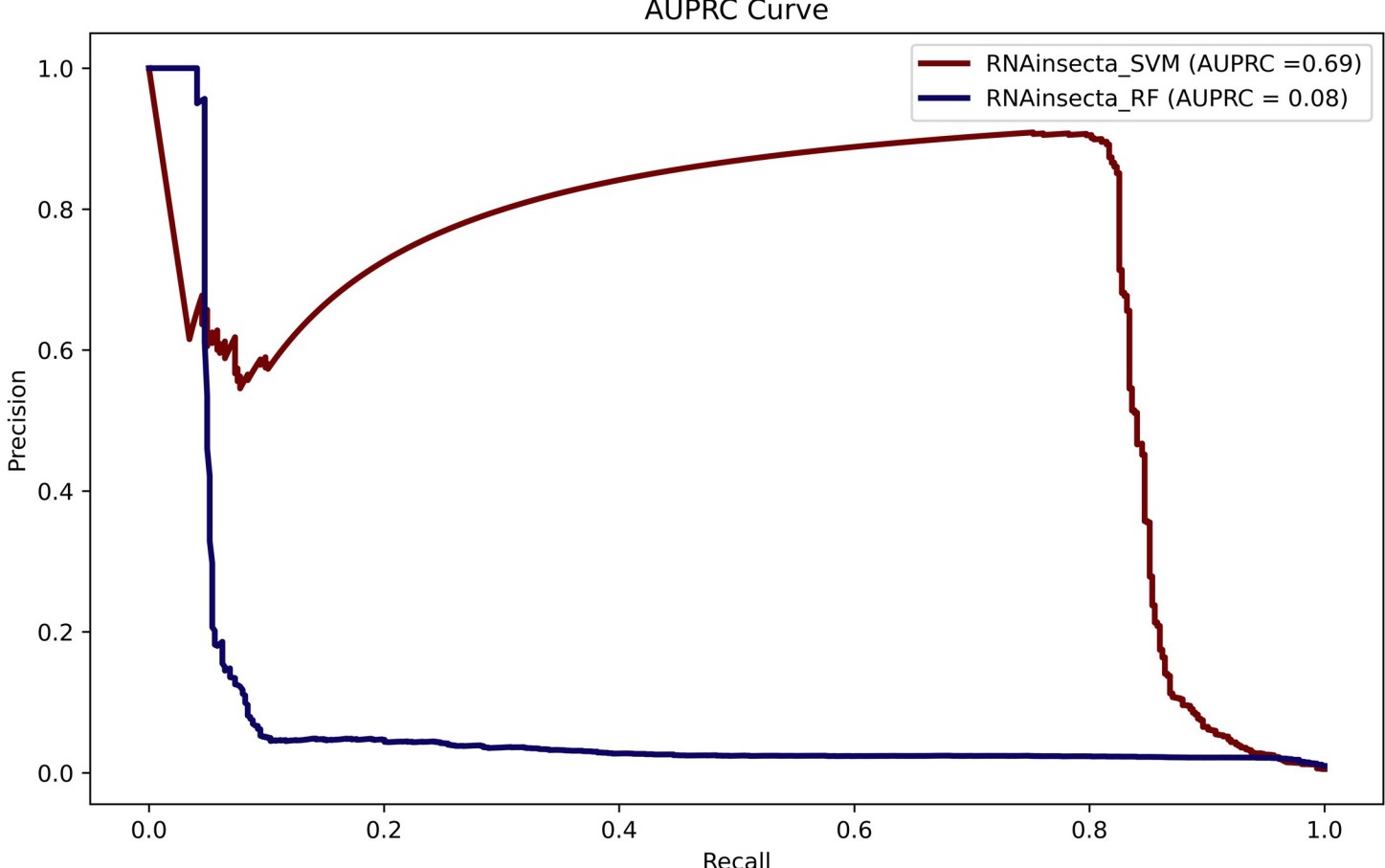

**Fig 3. AUPRC of the ML models.**

cleavage site of pre-miRNA to give mature miRNA is dynamically created depending upon the occurrence and position of the hairpin loops predicted in the secondary structure on the selected 3' or 5' directionality. User defined miRNA sequence can also be provided as query in case user already has mature miRNA sequence from the pre-miRNA (Fig 4C.i). The result page contains a hyperlink to the corresponding FlyBase Transcript ID. It displays the gene with a probable target match. It also includes a hyperlink to the miRBase ID of the miRNA that has been experimentally found to control the particular FlyBase Transcript (Fig 4C.ii).

## Discussion

### Dataset

A total of 28 organisms were considered for the study out of which *B. mori* had 427 pre-miRNA which was the highest among all. A large number of pre-miRNAs belonged to the *Drosophila* genus. The consideration of large negative datasets made it a typical imbalanced dataset classification problem since the positive to negative class ratio was approximately 1:6. The problem with such classification is that the majority of data considered for the classification belongs to a single class and hence the results are misleading.

**Table 6. Performance of RNAinsecta_RF in comparison with miPred for prediction of pre-miRNA across related phyla.** pre-miRNA of different species from Nematoda, Platyhelminthes, Virus and Mollusca and their performance based on TP and SN is shown.

| Phylum | Species | Total | RNAinsecta_RF | | miPred | |
|---|---|---|---|---|---|---|
| | | | TP | SN | TP | SN |
| **Nematoda** | *Brugia malayi* | 157 | 122 | 0.7770 | 119 | 0.7579 |
| | *Caenorhabditis brenneri* | 214 | 208 | 0.9719 | 194 | 0.9065 |
| | *Caenorhabditis elegans* | 253 | 207 | 0.8181 | 209 | 0.8260 |
| | *Ascaris suum* | 97 | 82 | 0.8453 | 86 | 0.8865 |
| | *Pristionchus pacificus* | 353 | 302 | 0.8555 | 307 | 0.8696 |
| **Platyhelminthes** | *Fasciola hepatica* | 38 | 27 | 0.7105 | 25 | 0.6578 |
| | *Gyrodactylus salaris* | 60 | 52 | 0.8667 | 43 | 0.7166 |
| | *Schistosoma mansoni* | 115 | 86 | 0.7478 | 53 | 0.4609 |
| | *Echinococcus granulosus* | 111 | 72 | 0.6486 | 81 | 0.7298 |
| | *Schmidtea mediterranea* | 148 | 126 | 0.8514 | 114 | 0.7703 |
| **Virus** | *Duck enteritis* | 24 | 19 | 0.7917 | 13 | 0.5417 |
| | *Epstein barr* | 25 | 21 | 0.84 | 23 | 0.92 |
| | *Human cytomegalovirus* | 15 | 10 | 0.6667 | 9 | 0.6 |
| | *Mouse cytomegalovirus* | 18 | 12 | 0.6667 | 12 | 0.6667 |
| **Mollusca** | *Lottia gigantea* | 59 | 46 | 0.7797 | 55 | 0.9322 |
| | *Melibe leonina* | 90 | 83 | 0.9222 | 85 | 0.9444 |

## Performance evaluation

CV score helps in the initial choice of the hyperparameters, however, to regularize the classifiers which performed well, they were tested on X_test that was initially kept separate. The NM models did not perform well on unseen data as was expected from their CV scores. The accuracy of the NM classifiers dropped quite below their CV Scores. As the training data of NM had lesser negative class, the models could not learn to classify non-miRNA which closely resemble true miRNAs and hence suffered from Type I error. The SN of these models were quite high suggesting they learned fairly well to classify positive miRNAs but produced a lot of FP as their SP was low. Hence, these models had poor precision and accuracy for which they were discarded from further analysis. Also, the 8494 human_CDS performed extremely poor with X_test and hence, it was discarded from 10-Fold CV parameter optimization.

The SMOTE models performed well on the test data as the models learned to correctly classify non-miRNAs which included insect CDS hairpins that closely resembled true miRNAs. The accuracy of Logistic Regression was the lowest among all the SMOTE models but still had higher MCC and $F_1$ than the KNN model. The MCC of SVM and RF were the highest among all the models.

**Table 7. No. of targets from each chromosome of *Drosophila melanogaster*.**

| Chromosome | No. of Sequence |
|---|---|
| 2L | 34 |
| 2R | 7 |
| 3L | 46 |
| 3R | 55 |
| X | 23 |
| 4 | 9 |

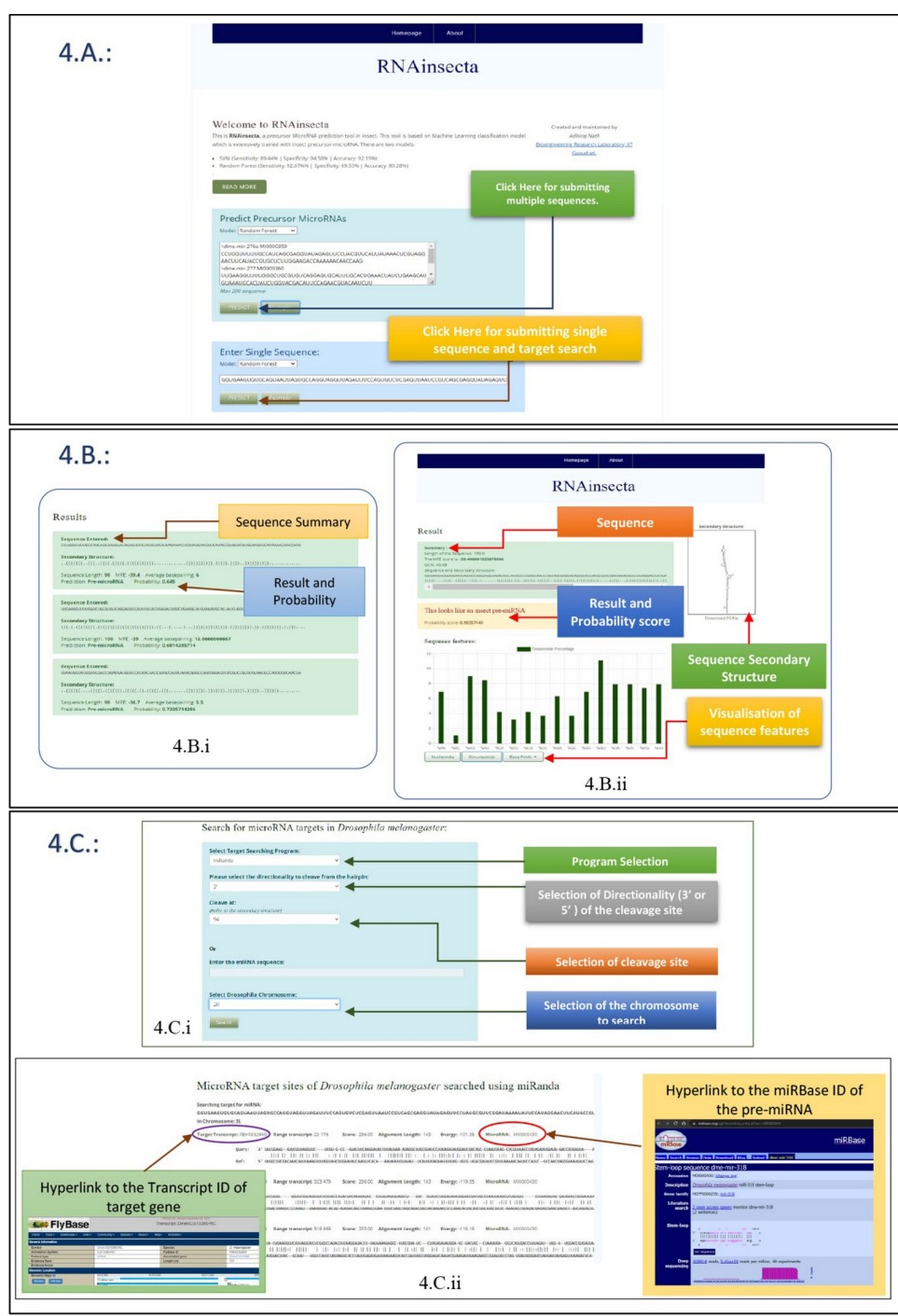

**Fig 4. User Interface of RNAinsecta web server.** A. shows the homepage of RNAinsecta containing space for both batch and single sequence query. B. is the result page of the query. B.i. and .B.ii. show output for batch and single sequence query respectively. C.i. contains the user interface for searching miRNA targets in *Drosophila melanogaster*. C.ii. shows the result of miR target search containing Transcript ID and its hyperlink to FlyBase as well as miRBase ID and its hyperlink.

The performance of SMOTE was better than NM suggesting that with increase in the amount of training data, the performance of these classifiers improves. Hence, for the

validation test the SMOTE models were selected for their performance analysis and comparison with already existing tools.

Although the imbalance dataset models performed extremely well with X_test, their performance drastically dropped with V_test. The models failed to produce any proper prediction, showing it to be a typical case of overfitting which was expected due to class imbalance. The models over learned from the majority class and classified every sequence as negative pre-miRNA. Hence, it indicated that our initial assumption to balance the dataset was necessary.

The outcome of M_test suggest that the model has good specificity, sensitivity and accuracy but since the ratio of positive to negative class was huge, hence the MCC recall and precision dropped significantly. However, the usability of the tool is not RNA-Seq data analysis but rather PCR products or small-scale synthesis of pre-miRNA in which respect the works fairly well. We believe the RNAinsectaSVM model is better suited for the prediction. While RNAinsectaSVM is available on the web, we have not removed RNAinsectaRF based on the precision and recall of M_test since it is still a better estimator than the other available tool which share the same PCR based methodology for detection as discussed below.

## Comparison with previous tools

Tools such as microPred, miPred, Triplet-SVM and MiPred are trained on the 8494 human pseudo pre-miRNA sequence as the negative dataset. HuntMi on the other hand uses many classes of CDS such as plant, virus, human, arabidopsis and along with 8494 human pseudo pre-miRNA as their negative dataset and have different classifiers for them. However, it does not contain any insect specific classifier. In our study we used the negative dataset that closely resembled with true insect pre-miRNA. Hence, most of the tools classified them as true miRNA making the Type I error. HuntMi and MiPred had the least Specificity with 0.31 and 0.13 respectively. HuntMI had an $F_1$ score of 70.44% but precision was 55.03%. microPred although had 71.37% accuracy, the specificity was 57%. Triplet-SVM and miPred performed well with MCC of 62.57% and 65.4% respectively, which was the highest among the previously developed tools considered for this experiment.

Triplet-SVM is trained solely on SVM classifier, with the 32 triplicate features. There is no mention of CV optimization of their hyperparameters. [20] MiPred is exclusively trained on Random Forest with the 32 triplet-SVM features along with dinucleotide shuffling and p-value of randomization [21]. miPred uses SVM RBF kernel with nucleotide thermodynamics features [27]. microPred uses 29 features from miPred and along with 12 modified features and is trained only on SVM classifier [35]. HuntMi uses 21 feaatures from microPred and uses 7 additional features such as loop length, orf, etc. [34]. These tools are based on command-line interface without UI/UX support. The provision for target prediction is not available is these tools. In our approach, we trained 4 datasets, on 93 features with 4 different ml algorithms and have also provided provisions for further analysis of the miRNA targets.

The SMOTE trained models of RF and SVM in our experiment had fairly good sensitivity but the Logistic Regression and KNN model suffered from the same Type I error. The RF model had accuracy and precision of 80.28% and 81.36% which was higher than all the previous tools tested on V_test. However, the best performance was given by the SVM model with specificity of 94.58% which was the highest among all models used in the experiment indicating it had the least Type I error. The accuracy, precision and $F_1$ score of the SVM model was also highest with 92.19%, 93.47% and 91.41% respectively. However, to achieve such low FP the model was allowed to make few Type II errors for which sensitivity of the model was lower than RF but yet was more than Triplet-SVM and miPred. The MCC score of SVM was 85.48% which was found to be the highest. As RF and SVM models performed better than all the

models, both were considered for implementation in a web server called RNAinsecta and the choice of model to select will depend on the user's requirement of specificity in their experiment.

**ROC.**   Triplet-SVM and miPred have lesser FPR than RNAinsecta_RF model but more than RNAinsecta_SVM. Tools like microPred and HuntMi although have high TPR also have high FPR for which their AUC is less. RNAinsecta SVM and RF had the highest AUC with 0.92 and 0.83 respectively followed by miPred and Triplet-SVM with 0.78 and 0.75 respectively. The smoothness of the curve is due to using binomial values instead of probability values to maintain uniformity among all the tools, as most of the them do not provide probabiity values.

**Performance on other phyla.**   RNAinsecta_RF performed well on Nematoda with highest prediction specificity. The performance on Platyhelminthes was better as compared to miPred. The performance on Virus was almost same as miPred whereas in case of Mollusc, miPred performed better.

**RNAinsecta webserver.**   RNAinsecta is currently hosted at https://rnainsecta.in/. The user-interface (UI) has both batch and single sequence searching modes. The batch mode allows maximum 200 FASTA sequences as input. After initial screening, user may check the single sequence search mode for further obtaining the mature miRNA and searching its targets. Users need to first select the program they want to use from the drop-down menu, the default selection is miRanda. The users then can select the orientation of the miRNA to be cleaved from pre-miRNA to be either 3' or 5' as both directionality strands are found to regulate the target gene [63]. In case of more than one hairpin, user can choose the cleavage site based on the secondary structure. Finally, the chromosome of *Drosophila* has to be selected where the target has to be searched. The resulting window will contain the list of possible targets genes and their FlyBase ID along with the miRBase ID of the miRNA that has been reported to regulate that gene. We have given three programs for the prediction of targets so that the users can select the one that best fit their requirement.

## Conclusion

In this work we present a new web-based tool for predicting novel pre-miRNA in insects and also search for their targets. We used 93 features sequence and thermodynamic characteristics of pre-miRNA. These features were trained on various ML algorithms such as SVM, Random Forest, Logistic Regression and KNN for binary classification of true and pseudo pre-miRNA. SMOTE and Near-Miss were used to handle the imbalance in the class, along with 10-fold cross-validation. Two models were selected upon their performance evaluation based on SVM and RF with accuracy of 92.19% and 80.28% respectively, tested on independent validation dataset along with other previous tools.

Further, the target for candidate miRNA produced from the pre-miRNA can be searched for the known miRNA regulated genes in *Drosophila melanogaster*. The target regions are the genes which are known to be regulated by miRNAs and therefore the user can check the details about the gene from the provided hyperlink.

To our knowledge this is the first tool which provides prediction of insect pre-miRNA as well as target searching for the resulting miRNA.

In the future, this tool can be used to predict pre-miRNA in a given transcriptome by implementing the source to RNA-Seq data analysis pipeline. In this way, it will help to find the collection and abundance of pre-miRNA in a given condition.

## Supporting information

**S1 File. Comparative analysis of pre-miRNA features of insects with other organisms.**
(DOCX)

**S2 File. The f score and *p*-value of all 93 features of positive and negative set used for training.**
(CSV)

**S3 File. Cross validation score for each fold during training and parameter optimisation.**
(XLSX)

## Acknowledgments

We are thankful to team Param-Ishan at IIT Guwahati for letting us train our models in HPC cluster. Author AN expresses gratitude towards MHRD (Ministry of Human Resource Development) and IITG (Indian Institute of Technology Guwahati) for financial support in the form of scholarship.

We are extremely grateful to the reviewers for their valuable comments and feedbacks that helped to improve the manuscript significantly.

## Author Contributions

**Conceptualization:** Utpal Bora.

**Methodology:** Adhiraj Nath.

**Project administration:** Adhiraj Nath.

**Resources:** Adhiraj Nath.

**Software:** Adhiraj Nath.

**Supervision:** Utpal Bora.

**Validation:** Adhiraj Nath.

**Writing – original draft:** Adhiraj Nath.

**Writing – review & editing:** Utpal Bora.

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
