## [Decision Letter · Decision Letter 0]

13 Apr 2023

PONE-D-23-06543**RNAinsecta:** A tool for prediction of precursor microRNA in insects and search for their target in the model organism * Drosophila melanogaster*.PLOS ONE

Dear Dr. Bora,

Thank you for submitting your manuscript to PLOS ONE. After careful consideration, we feel that it has merit but does not fully meet PLOS ONE’s publication criteria as it currently stands. Therefore, we invite you to submit a revised version of the manuscript that addresses the points raised during the review process.

We look forward to receiving your revised manuscript.

Kind regards,

Abu Sayed Chowdhury, Ph.D.

Academic Editor

PLOS ONE

Journal Requirements:

   "This work is not funded by any body. The funders had no role in study design, data collection and analysis, decision to publish, or preparation of the manuscript."

   "N/A" 

Additional Editor Comments:

Upon assessment by the reviewers, it was advised that major revisions are needed for the manuscript. Queries have been raised by the reviewers concerning your dataset, methodologies, and findings. Please review the comments attached herewith. If you are able to fully address the reviewers' concerns and make improvements to the manuscript, please submit the revised version.

Reviewers' comments:

Reviewer's Responses to Questions

**Comments to the Author**

1. Is the manuscript technically sound, and do the data support the conclusions?

Reviewer #1: Partly

Reviewer #2: Yes

Reviewer #3: Yes

2. Has the statistical analysis been performed appropriately and rigorously? 

Reviewer #1: Yes

Reviewer #2: N/A

Reviewer #3: No

3. Have the authors made all data underlying the findings in their manuscript fully available?

Reviewer #1: Yes

Reviewer #2: Yes

Reviewer #3: Yes

4. Is the manuscript presented in an intelligible fashion and written in standard English?

Reviewer #1: Yes

Reviewer #2: Yes

Reviewer #3: Yes

5. Review Comments to the Author

Reviewer #1: Authors present a study of models for pre-miRNA identification specifically on insects. I found it to be a well-written and informative, and I believe that your tool will be of great use to researchers working with insects. I also found it resourceful the use of miRNA target prediction as a proxy to detect possible pre-miRNAs.

However, I have a couple of points that I think could be further addressed.

- Firstly, it is mentioned that there are deep learning-based methods such as mire2e for predicting precursor microRNAs. Have you tried implementing these methods in your framework and compared their performance with your current approach? I do not intend to add too much burden to the already extensive tests performed for this work, but there are lots of recent tools to explore pre-miRNAs that improved the ones mentioned in the tables.

- I understand that the high reported accuracy is result of a balanced test set. While this is useful for assessing the recall of the model, it may not give a complete picture of the precision of the tool. Have you considered how will the model be used? for example, when searching thousands of sequences in search of pre-miRNAs, which will be the false positives rates?

- There are tools to assess if model have a hairpin-like structure. This may be helpful to improve negative dataset, as both human pre-miRNAs and coding RNA are very different to the actual pre-miRNAs.

Overall, I think RNAinsecta makes a valuable contribution to the field, so I hope this suggestions helps to improve the manuscript.

Reviewer #2: In this study, Nath et al. developed a novel tool for specially predicting precursor microRNA of insects. This study is very solid and the tool developed deserves further consideration for the related researchers. Although many details have been considered well, some obvious defects should be addressed in detail. (1) For example, as everyone knows, the performance of prediction is dependent on construction of negative samples. The more difference between positive and negative samples, the better of performance. The authors should give an intuitive comparison between positive and negative samples. (2) To further improve the accuracy of target prediction, the more tools for target prediction should be used. Then, the intersection of predicted results from tools are provided back to users. (3) In general, the ROC curve is very smooth. But as depicted in Fig. 2, the authors should clarify why. Suggest the authors use more datasets to do comparison of performance between different tools.

Reviewer #3: The authors of the manuscript entitled “RNAinsecta: A tool for prediction of precursor microRNA in insects and search for their target in the model organism Drosophila melanogaster” describe a microRNA discovery model for insects. They make use of and combine previously reported features to train various models on known insect miRNA and derived pseudo insect miRNA. The authors briefly discuss how microRNAs in insects have distinct differences from human, mouse, and plant microRNAs, specifically, MFE and GC%. They report that due to these differences, published methods perform poorly on insects. The authors put forth two models from their experimentation, an SVM and a random forest model trained on a SMOTE-balanced dataset. The model’s hyperparameters were tuned using random search, grid search, and 10-fold cross validation.

The authors clearly describe the features used and their methodology for hyperparameter tunning. However, some clarification can especially to the introduction and discussion to further improve the understanding of the manuscript.

Major points:

- The authors estimate the generalizability of their classifiers on an artificially “balanced” dataset. In practise the ratio of positive:negative miRNA would not be balanced. In fact, the class imbalance for a sequence-based miRNA discovery method can be as high as 1000 negatives for each positive miRNA, when considering all hairpins in a genome that look like a pre-miRNA. The authors should report performance on a “naturally imbalanced” test set, reflecting the realistic deployment of the predictor to an entire insect genome. Prevalence-corrected precision can be used to estimate performance at different class imbalance levels, for example.

- The website predicts possible targets for the miRNA inputted by users; however, the prediction seems to be made between the pre-miRNA and mRNA. It has been reported in literature that mature miRNA target mRNA, and not pre-miRNA.

- The ROC curves reported appear to be incorrect since they only contain three points (bottom left, some mid-point, and top-right). Instead, ROC curves should be more smooth, illustrating the achievable TPR and FPR for many different decision thresholds. Perhaps the Python sklearn “predict” function was used to predict binary classes “0” or “1” instead of “predict_proba” that produces a prediction confidence between 0 and 1. The graphs should be recomputed along with summary statistics, such as AUC-ROC.

Minor points:

- The authours should review the consistency of acronyms in the document. For example, “pre-mirna” and “pre-miRNA” are both present in the manuscript.

- It is not clear in the methodology if the SMOTE and/or NM was applied to the X_test or V_test datasets. Considering that the application of those methods would constitute a methodological issue, it would be best if that distinction was made abundantly clear. This relates to the major issue of testing on “artificially balanced” test sets above.

- The authors consider many metrics to estimate performance, most of which are not wholly suitable for representing model performance in the presence of class imbalance. The authors do consider MCC, but few discussions are made on those results. Additional discussion of the different models performance based on MCC, F1 measure, or prevalence-corrected precision should be added to the manuscript

o Additionally, the authors should consider reporting Area under the precision recall curve (AUPRC), when also using the prevalence-corrected precision, as it is representative of the performance of classifiers on “naturally imbalanced” datasets.

- The authors report the performance of a KNN model architecture among others. Was a consideration made to normalize the features as it has a significant effect on the performance of KNN model, especially if the feature has a variety of ranges?

- It would be beneficial in the introduction to include more discussion of the difference between insect miRNA biogenesis and animal/plant miRNA biogenesis. Especially, since the foundation of the motivation of the manuscript is that miRNAs from insects are very different from animal and plant miRNA and thus require their own microRNA discovery predictor.

6. PLOS authors have the option to publish the peer review history of their article (what does this mean?). If published, this will include your full peer review and any attached files.

Reviewer #1: **Yes: **Leandro Bugnon

Reviewer #2: No

Reviewer #3: No

---

## [Author Response · Author response to Decision Letter 0]

8 May 2023

To

The Editor

PLOS One

Sub: Revision of the manuscript Entitled “RNAinsecta: A tool for prediction of precursor microRNA in insects and search for their target in the model organism Drosophila melanogaster”.

Dear Sir,

We express our sincere gratitude for consideration of our manuscript for peer-reviewing. We thank the reviewers for evaluation of our manuscript and letting us know the areas which required improvements. We have made significant changes to the manuscript based on the valuable feedback provided by all three esteemed reviewers. We have addressed most, if not all, of their suggestions. 

Summary of reviews and changes:

We have made significant changes in the manuscript from the valuable feedbacks provided by all three esteemed Reviewers. We have provided an imbalance test with large negative samples as suggested by all the Reviewers. We have shown the result of this analysis as AUPRC plot as suggested by Reviewer 3 and have also provided a table for other performance matrices. 

In response to Reviewer 2's suggestion, we have added two additional tools (miRmap and RNAhybrid) for target prediction, which can be selected from the drop-down menu provided in the miRNA pre-processing form. We have also provided explanations regarding the smooth ROC and addressed other minor issues in the manuscript. We have addressed each major and minor comment provided by Reviewer 3.

 

Responses to Reviewer 1 comments:

Reviewer #1: Authors present a study of models for pre-miRNA identification specifically on insects. I found it to be a well-written and informative, and I believe that your tool will be of great use to researchers working with insects. I also found it resourceful the use of miRNA target prediction as a proxy to detect possible pre-miRNAs.

However, I have a couple of points that I think could be further addressed.

1. Firstly, it is mentioned that there are deep learning-based methods such as mire2e for predicting precursor microRNAs. Have you tried implementing these methods in your framework and compared their performance with your current approach? I do not intend to add too much burden to the already extensive tests performed for this work, but there are lots of recent tools to explore pre-miRNAs that improved the ones mentioned in the tables.

We thank esteemed Reviewer for the suggestion. We had initially tried to use mire2e. Correct us if we are wrong but the philosophy of mire2e is considerably different than our tool as in mire2e the expected sequence is typically a long chromosome which are concatenated into pre-miRNAs. Due to this reason, as pre-miRNA of insects are longer, they get cleaved and thus, there are lot of false negatives. 

2. I understand that the high reported accuracy is result of a balanced test set. While this is useful for assessing the recall of the model, it may not give a complete picture of the precision of the tool. Have you considered how will the model be used? for example, when searching thousands of sequences in search of pre-miRNAs, which will be the false positives rates?

We have worked on this suggestion as recommended by all the reviewers and have taken a larger dataset for comparison. As in a typical cell, most coding mRNAs are longer in length, hence we selected only those non-coding hairpins that closely resemble to pre-miRNA but are not. In this way we have included the new analysis and have given an AUPRC plot to give the precision recall curve as happens with large imbalanced data. (line 432, 554)

3. There are tools to assess if model have a hairpin-like structure. This may be helpful to improve negative dataset, as both human pre-miRNAs and coding RNA are very different to the actual pre-miRNAs.

The hairpin like structure can be inferred from the secondary structure of the pre-miRNA. We have already taken care of that during our feature calculation in the backend server script. Any sequence that does not form a hairpin loop is automatically discarded with the help of pattern matching using regular expression.

Overall, I think RNAinsecta makes a valuable contribution to the field, so I hope this suggestions helps to improve the manuscript.

We feel honoured and privileged by such kind and insightful feedbacks.

 

Responses to Reviewer 2 comments:

Reviewer #2: In this study, Nath et al. developed a novel tool for specially predicting precursor microRNA of insects. This study is very solid and the tool developed deserves further consideration for the related researchers. Although many details have been considered well, some obvious defects should be addressed in detail. 

(1) For example, as everyone knows, the performance of prediction is dependent on construction of negative samples. The more difference between positive and negative samples, the better of performance. The authors should give an intuitive comparison between positive and negative samples. 

We have worked on this suggestion and have taken a larger dataset for comparison. As in a typical cell, most coding mRNAs are longer in length, hence we selected only those non-coding hairpins that closely resemble to pre-miRNA but are not. In this way we have included the new analysis and have given an AUPRC plot to give the recall value as happens with large imbalanced data. (line 432, 554)

(2) To further improve the accuracy of target prediction, the more tools for target prediction should be used. Then, the intersection of predicted results from tools are provided back to users. 

We thank the esteemed reviewer for this suggestion. We have implemented two additional target prediction tools namely miRmap and RNAhybrid along with miRanda. The users can now select the choice of program from the dropdown menu and provide the remaining necessary parameters to search for their miRNA target in Drosophila m. transcripts. 

(3) In general, the ROC curve is very smooth. But as depicted in Fig. 2, the authors should clarify why. Suggest the authors use more datasets to do comparison of performance between different tools.

We understand that our ROC does not look like a regular one which is due to the output as binary values instead of probability scores. The previous tools do not give output in probability values but in 0s and 1s (yes or no) and to maintain uniformity in the plot, we also used the same. This type of plot for pre-miRNA detection has been reported in other tools such as tamiRpred. We added this clarification in the main text as well in ROC section (line 419).

 

Responses to Reviewer 3 comments:

Reviewer #3: The authors of the manuscript entitled “RNAinsecta: A tool for prediction of precursor microRNA in insects and search for their target in the model organism Drosophila melanogaster” describe a microRNA discovery model for insects. They make use of and combine previously reported features to train various models on known insect miRNA and derived pseudo insect miRNA. The authors briefly discuss how microRNAs in insects have distinct differences from human, mouse, and plant microRNAs, specifically, MFE and GC%. They report that due to these differences, published methods perform poorly on insects. The authors put forth two models from their experimentation, an SVM and a random forest model trained on a SMOTE-balanced dataset. The model’s hyperparameters were tuned using random search, grid search, and 10-fold cross validation.

The authors clearly describe the features used and their methodology for hyperparameter tunning. However, some clarification can especially to the introduction and discussion to further improve the understanding of the manuscript.

Major points:

1. The authors estimate the generalizability of their classifiers on an artificially “balanced” dataset. In practise the ratio of positive:negative miRNA would not be balanced. In fact, the class imbalance for a sequence-based miRNA discovery method can be as high as 1000 negatives for each positive miRNA, when considering all hairpins in a genome that look like a pre-miRNA. The authors should report performance on a “naturally imbalanced” test set, reflecting the realistic deployment of the predictor to an entire insect genome. Prevalence-corrected precision can be used to estimate performance at different class imbalance levels, for example.

We thank the esteemed reviewer for this suggestion. In case of naturally balanced dataset, the RNA transcripts contain long mRNA sequences which can easily be classified negative. Hence we selected only those non-coding hairpins that closely resemble to pre-miRNA but are not. In this way we have included the new analysis and have given an AUPRC plot to give the recall value as happens with large imbalanced data. (line 432, 554)

2. The website predicts possible targets for the miRNA inputted by users; however, the prediction seems to be made between the pre-miRNA and mRNA. It has been reported in literature that mature miRNA target mRNA, and not pre-miRNA.

We are sorry for this confusion. We had previously mentioned it in Web Application and miRNA Targets of the Result section at line 500 about the pre-processing of pre-miRNA to miRNA where we used regular expression in JavaScript to select portions from pre-miRNA that can possibly form miRNA, and also the users can provide their own sequence for better results.

3. The ROC curves reported appear to be incorrect since they only contain three points (bottom left, some mid-point, and top-right). Instead, ROC curves should be more smooth, illustrating the achievable TPR and FPR for many different decision thresholds. Perhaps the Python sklearn “predict” function was used to predict binary classes “0” or “1” instead of “predict_proba” that produces a prediction confidence between 0 and 1. The graphs should be recomputed along with summary statistics, such as AUC-ROC.

We understand that our ROC does not look like a regular one which is due to the output as binary values instead of probability scores. The previous tools do not give output in probability values but in 0s and 1s (yes or no) and to maintain uniformity in the plot, we also used the same. This type of plot for pre-miRNA detection has been reported in other tools such as tamiRpred. We added this clarification in the main text as well in ROC section (line 419).

Minor points:

1. The authours should review the consistency of acronyms in the document. For example, “pre-mirna” and “pre-miRNA” are both present in the manuscript.

We thank the reviewer for pointing it out. We have now corrected it in our manuscript.

2. It is not clear in the methodology if the SMOTE and/or NM was applied to the X_test or V_test datasets. Considering that the application of those methods would constitute a methodological issue, it would be best if that distinction was made abundantly clear. This relates to the major issue of testing on “artificially balanced” test sets above.

We are sorry that it is not clear. We had mentioned it previously in line 216 of the “Test Set” section that these datasets were not artificially balanced. We have now explained it in some more detail.

3. The authors consider many metrics to estimate performance, most of which are not wholly suitable for representing model performance in the presence of class imbalance. The authors do consider MCC, but few discussions are made on those results. Additional discussion of the different models performance based on MCC, F1 measure, or prevalence-corrected precision should be added to the manuscript

o Additionally, the authors should consider reporting Area under the precision recall curve (AUPRC), when also using the prevalence-corrected precision, as it is representative of the performance of classifiers on “naturally imbalanced” datasets.

We thank the esteemed reviewer for this suggestion. We have additionally provided and account of AUPRC for imbalance dataset as described previously along with some detail explanation of performance matrices.

4. The authors report the performance of a KNN model architecture among others. Was a consideration made to normalize the features as it has a significant effect on the performance of KNN model, especially if the feature has a variety of ranges?

Yes, we used standard scaler for these normalizations. 

5. It would be beneficial in the introduction to include more discussion of the difference between insect miRNA biogenesis and animal/plant miRNA biogenesis. Especially, since the foundation of the motivation of the manuscript is that miRNAs from insects are very different from animal and plant miRNA and thus require their own microRNA discovery predictor.

We are thankful for the suggestion. We have added more details on the biogenesis in pre-miRNA in insects with respect to other organisms.

---

## [Decision Letter · Decision Letter 1]

4 Jun 2023

**RNAinsecta:** A tool for prediction of precursor microRNA in insects and search for their target in the model organism * Drosophila melanogaster*.

PONE-D-23-06543R1

Dear Dr. Bora,

We’re pleased to inform you that your manuscript has been judged scientifically suitable for publication and will be formally accepted for publication once it meets all outstanding technical requirements.

Kind regards,

Abu Sayed Chowdhury, Ph.D.

Academic Editor

PLOS ONE

Additional Editor Comments (optional):

Thank you for addressing reviewers' questions, comments, and suggestions.

Reviewers' comments:

Reviewer's Responses to Questions

**Comments to the Author**

1. If the authors have adequately addressed your comments raised in a previous round of review and you feel that this manuscript is now acceptable for publication, you may indicate that here to bypass the “Comments to the Author” section, enter your conflict of interest statement in the “Confidential to Editor” section, and submit your "Accept" recommendation.

Reviewer #1: All comments have been addressed

Reviewer #2: All comments have been addressed

Reviewer #3: All comments have been addressed

2. Is the manuscript technically sound, and do the data support the conclusions?

Reviewer #1: Yes

Reviewer #2: Yes

Reviewer #3: Yes

3. Has the statistical analysis been performed appropriately and rigorously? 

Reviewer #1: Yes

Reviewer #2: Yes

Reviewer #3: Yes

4. Have the authors made all data underlying the findings in their manuscript fully available?

Reviewer #1: Yes

Reviewer #2: Yes

Reviewer #3: Yes

5. Is the manuscript presented in an intelligible fashion and written in standard English?

Reviewer #1: Yes

Reviewer #2: Yes

Reviewer #3: Yes

6. Review Comments to the Author

Reviewer #1: All my suggestions were addresed. As suggested by the authors, it may require to adjust parameters of mire2e to adapt to this kind of pre-miRNAs (the model has to be retrained and evaluated, it is a complete new work).

Great work!

Reviewer #2: (No Response)

Reviewer #3: The authors have addressed all my comments adequately. It is unfortunate that they have chosen not to illustrate their model's performance using ROC curves, even if the methods against which they are comparing have chosen to only output binary prediction scores. Continuous prediction scores (i.e., prediction and confidence indicator) can be used to create ROC curves and provide valuable information for potential users.

7. PLOS authors have the option to publish the peer review history of their article (what does this mean?). If published, this will include your full peer review and any attached files.

Reviewer #1: **Yes: **Leandro Bugnon

Reviewer #2: No

Reviewer #3: No

---

## [Editor Report · Acceptance letter]

6 Jun 2023

PONE-D-23-06543R1 

RNAinsecta: A tool for prediction of precursor microRNA in insects and search for their target in the model organism * Drosophila melanogaster*. 

Dear Dr. Bora:

I'm pleased to inform you that your manuscript has been deemed suitable for publication in PLOS ONE. Congratulations! Your manuscript is now with our production department. 

Kind regards, 

on behalf of

Dr. Abu Sayed Chowdhury 

Academic Editor

PLOS ONE